# Preparation of Key Intermediates for the Syntheses of Coenzyme Q_10_ and Derivatives by Cross-Metathesis Reactions

**DOI:** 10.3390/molecules25030448

**Published:** 2020-01-21

**Authors:** Trang Nguyen, Hung Mac, Phong Pham

**Affiliations:** 1Laboratory of Catalysis at Faculty of Chemistry, VNU University of Science, Vietnam National University, Hanoi 110403, Vietnam; nguyentrang2484@gmail.com; 2Laboratory of Medicinal Chemistry of Faculty of Chemistry, VNU University of Science, Vietnam National University, Hanoi 110403, Vietnam

**Keywords:** coenzyme Q_10_, solanesol, claisen rearrangement, stille coupling, cross-metathesis.

## Abstract

An alternative catalytic strategy for the preparation of benzylmethacrylate esters, key intermediates in the synthesis of coenzyme Q_10_ and derivatives, was reported. This strategy avoided undesirable stoichiometric reduction/oxidation processes by utilizing the catalytic formation of allylarenes and then cross-metathesis to selectively form *E*-benzylmethacrylate esters with good yields (58–64%) and complete *E*-selectivity. The ester intermediates were reduced to common key benzylallylic alcohols (90–92% yield), which were subsequently used in the formal syntheses of coenzyme Q_10_ and one derivative.

## 1. Introduction

Coenzyme Q_10_ (**1**), an isoprenylated quinone, is a critical electron-transfer compound found in all respiring eukaryotic cells [1,2,3]. The demand for commercial coenzyme Q_10_ calls for efficient artificial syntheses of this compound [4,5,6,7,8,9,10,11]. The frame of coenzyme Q_10_ can be viewed as a combination of terpenyl and aromatic fragments. Therefore, the simplest terpenyl moiety from solanesol, found in tobacco leaves, and the common aromatic moiety benzylallyl alcohol (**2c**) were utilized to build up coenzyme Q_10_ molecule [11]. In particular, previous studies [11] have utilized a Friedel–Craft allylation of a quinol derivative with 2-methylbut-3-en-2-ol followed by a series of oxidation/reduction reactions to achieve the desired target (**2c**). Alcohol (**2c**) was then used in transformations with the solanesyl bromide obtained from the bromination of solanesol to achieve coenzyme Q_10_ (Scheme 1). This strategy served as the framework for the synthesis of coenzyme Q_10_ based on the terpenyl chain from solanesol.

However, the preparation of the *iso*-pentenylated quinol in the reported work unexpectedly required an additional reduction step. Re-examining the transformation, we found that quinone, the oxidized form of the desired *iso*-pentenylated quinol, was in fact the major product in an inseparable mixture of the two compounds. Indeed, upon exposure to air, all the amount of expected quinol was oxidized to its corresponding quinone after 45 min. A similar unexpected oxidation was also observed by the Hecht group [12]. This result led to an undesirable stoichiometric reduction step being added to the synthesis. The *iso*-pentenylated quinol was stoichiometrically oxidized by SeO_2_ to aldehyde, not alcohol, and therefore required an extra reduction step to obtain the benzylallyl alcohol (**2c**). The whole procedure to prepare the aromatic moiety was, therefore, accompanied by unenviable stoichiometric oxidation/reduction reactions. With the advantages of known catalytic processes, we anticipated that suitable catalytic transformations such as Claisen rearrangement, Stille coupling, and cross-metathesis could offer additional strategies and allow for catalytic access of coenzyme Q_10_ and derivatives.

### Design Plan

We hypothesized that extra reduction/oxidation steps for preparing benzylallyl alcohol (**2c**) quinol were required because the previous research used readily oxidizable quinol, and unfunctionalized *iso*-pentenyl group. We envisioned that allyarene (**7**) (Scheme 2) could be prepared from commercially available aryl bromides (**3**) by Stille coupling. This allylarene could then undergo a cross-metathesis reaction with the methyl methacrylate to offer the desired *E*-cross-metathesis products [13,14,15,16], which could be further reduced to the key synthetic intermediate benzyl alcohols (**2**). This catalytic strategy would allow for alternative access to common key intermediates in the syntheses of coenzyme Q_10_ and derivatives.

## 2. Results and Discussion

To examine the proposed idea, five allylated arenes (**7a**–**e**) were prepared. These allylarenes were synthesized either from commercially available aryl bromide **3** (path **A**) [17,18,19] or monoacetylquinol **4** (path **B**) [20,21,22,23,24]. (Scheme 3) The Stille coupling of corresponding (MeO)_2_ aryl bromides with allyltributylstannane directly afforded allylarenes **7b**–**c** and **7e** with high yields (75%–78%). Diacetate allylarenes **7a** and **7d** were prepared from corresponding quinol derivatives (**4**). First, quinols **4**(**a** and **d**) were allylated to provide allyl ethers **5a** (84%), and **5b** (82%), which then participated in the Claisen rearrangement to furnish allylated quinols **6a** (78%) and **6d** (55%). Allylated quinols **6a** and **6d** were then acetylated with Ac_2_O to afford **7a** (59% overall yield) and **7d** (41% overall yield).

Next, the cross-metathesis of allylated arenes with methyl methacrylate was studied. The results are shown in Table 1. Initially, the cross-metathesis reaction of **7a** was carried out in neat methyl methacrylate (Table 1, Entry 1). However, only a trace amount of the desired product was obtained. The reaction was then conducted in dichloromethane with a ratio of one equivalent of allylarene **7a** and five equivalents of methyl methacrylate. In this case, we mostly obtained the homo-coupling of allyarene (Table 1, Entry 2). Increasing the amount of methyl methacrylate to 20 equivalents, we were able to isolate desired the product **8a** with 56% yield and exclusive *E*-configuration based on NOESY NMR spectroscopy analysis. This result was consistent with the previous studies [13,14,15,16]. The reaction condition was also applied to allylarenes **7b**–**e**. All cross-metathesis reactions generated desired *E* isomers with moderate to good yields (**8b**–**e**, 58–65% yields) providing the key intermediates for the syntheses of coenzyme Q_10_ and derivatives (**8a**–**e**).

We then reduced esters **8b**,**c**,**e** by LiAlH_4_ to corresponding alcohols **2b**,**c**,**e** with very good yields (90%–92% yield) (Scheme 4). These alcohols could be used in well-established synthetic paths to furnish coenzyme Q_10_ [25] (from **2c**) and its derivatives.

### Synthesis of a Coenzyme Q_10_’s Derivative

Following a similar procedure for the formal synthesis of coenzyme Q_10_ [25], we obtained derivative **12** from the corresponding key alcohol (**2b**) (Scheme 5). First, the alcohol (**2b**) was converted into allyl bromide (**9**) (crude yield: 90%). The bromide (**9**, without purifying) was then treated with PhSO_2_Na to obtain the phenylsulfinate (**10**: 62%). The sulfinate (**10**) was then subjected to an allylation event with solanesyl bromide in the presence of *^t^*BuOK to form **11** with a moderate yield (58%). Finally, the de-sulfination of **11** afforded the coenzyme Q_10′_s derivativate (**12**: 43%)

## 3. Materials and Methods 

### 3.1. General Information

Commercial reagents were purified prior to use following the guidelines of Perrin and Armarego [26]. All solvents were redistilled before using. Reactions under N_2_ atmosphere were performed with a Schlenk apparatus or simple dry-box (Laboratory build, Hanoi, Vietnam). Organic solutions were concentrated under reduced pressure on a Büchi rotary evaporator using an ice-water bath for volatile compounds. Chromatographic purification of products was accomplished by flash chromatography on silica gel according to the method of Still [27]. Thin-layer chromatography (TLC) was performed on Merk silica gel plates. Visualization of the developed chromatogram was performed by fluorescence quenching, *p*–anisaldehyde, ceric ammonium molybdate, or potassium permanganate stains. ^1^H and ^13^C NMR spectra were recorded on a Bruker Advance–III 500 (500 and 125 MHz) instrument (Bruker Corp., Billerica, MA, USA) in the Faculty of Chemistry, VNU University of Science, Vietnam National University, Hanoi, Vietnam, and were internally referenced to residual protic solvent signals (note: CDCl_3_ referenced at d 7.27 and 77.0 ppm respectively). ^13^C NMR spectra were recorded in J-MOD (J-modulated spin-echo) mode, which were the combination of ^13^C and DEPT without losing quaternary ^13^C signals. Data for ^1^H NMR are reported as follows: chemical shift (δ ppm), multiplicity (s = singlet, d = doublet, t = triplet, q = quartet, h = heptet, m = multiplet, b = broad), integration, coupling constant (Hz), and assignment. Data for ^13^C NMR are reported in terms of chemical shift and multiplicity; no special nomenclature has been used for equivalent carbons. The IR spectrum (Appendix A) of alcohol **4** was recorded on the Shimadzu FTIR Affinity-1S (Shimadzu Corp., Kyoto, Japan) in the Department of Inorganic Chemistry, Faculty of Chemistry, VNU University of Science, Vietnam National University and has been reported in terms of frequency of absorption. High-resolution mass spectra were obtained via ESI method on Agilent 6530 Accurate-Mass Q-TOP LC/MS (Agilent Technonogy Inc., Santa Clara, CA, USA) in the Institute of Marine Biochemistry, Vietnam Academy of Science and Technology Hanoi, Vietnam.

### 3.2. General Procedure for Allylation Reaction 

NaH (0.78 g, 32.5 mmol) was slowly added to the solution of phenol (16 mmol) in DMF (60 mL) at 0 °C under an N_2_ atmosphere. The reaction mixture was stirred vigorously for 30 min to evaporate the gas in solution. After that, allyl bromide (2.18 mL, 25.2 mmol) was slowly added to the reaction flask and refluxed at 40 °C for 12 h. The mixture was poured into ice-water and extracted with ethyl acetate. The organic layer was washed with brine, dried over Na_2_SO_4_, and evaporated under vacuum. The residue was purified via flash column chromatography with silica gel as the stationary phase (0%–2% ethyl acetate/*n-*hexane) to yield the titular allyl aryl ether.

*4-(Allyloxy)-2,3,6-trimethylphenyl acetate* (**5a**): clear oil, 84% yield. IR (film) 2976, 2918, 2864, 1759, 1744, 1599, 1584, 1404, 1357, 1211, 1161, 1111, 1084, 937, 895, 839, 768, 745 cm^–1^; ^1^H NMR (500 MHz, CDCl_3_) δ 6.58 (s, 1H), 6.14–6.02 (m, 1H), 5.44 (dq, *J =* 17.3, 1.7 Hz,1H), 5.28 (dq, *J =* 10.5, 1.5 Hz, 1H), 4.50 (dt, *J =* 5.0, 1.6 Hz, 2H), 2.34 (s, 3H), 2.18 (s, 3H), 2.13 (s, 3H), 2.07 (s, 3H); ^13^C NMR (125 MHz, CDCl_3_) δ 169.5, 154.2, 141.8, 133.8, 129.8, 127.1, 124.3, 116.9, 111.6, 69.4, 20.6, 16.7, 13.1, 12.1; HRMS (ESI) exact mass calculated for [M + H]^+^ (C_14_H_19_O_3_) requires *m*/*z* 235.1334, found *m*/*z* 235.1329.

*4-(allyloxy)-2-methylnaphthalen-1-yl acetate* (**5d**): clear oil, 82% yield. IR (film) 3073, 2918, 2862, 1759, 1744, 1632, 1599, 1506, 1462, 1404, 1356, 1269, 1202, 1159, 1111, 1084, 937, 895, 839, 768, 745 cm^–1^; ^1^H NMR (500 MHz, CDCl_3_) δ 8.26 (d, J 8.3 Hz, 1H), 7.67 (d, *J =* 8.4 Hz, 1H), 7.52–7.48 (m, 1H), 7.45–7.42 (m, 1H), 6.65 (s, 1H), 6.20–6.14 (m, 1H), 5.53 (dd, *J* = 17.3, 1.5 Hz, 1H), 5.34 (dd, *J =* 6.8, 1 Hz, 1H), 4.70 (d, *J* 5.1 Hz, 2H), 2.46 (s, 3H), 2.30 (s, 3H); ^13^C NMR (125 MHz, CDCl_3_) δ 169.6, 152.3, 137.8, 133.2, 127.9, 127.2, 126.3, 124.9, 122.6, 120.6, 117.6, 107.7, 69.3, 20.7, 16.9; HRMS (ESI) exact mass calculated for [M + H]^+^ (C_16_H_17_O_3_) requires *m*/*z* 257.1178, found *m*/*z* 257.1172.

### 3.3. General Procedure for Claisen Rearrangement

A solution of allyl aryl ether (11 mmol) in 1,2-dichlorobenzene (50 mL) was stirred for 24 h at 180 °C under an N_2_ atmosphere. The mixture was purified by column chromatography with silica gel as the stationary phase (0%–5% ethyl acetate/*n-*hexane) to yield the titular allylphenol.

*3-allyl-4-hydroxy-2,5,6-trimethylphenyl acetate* (**6a**): yellow solid, 78% yield, mp = 119–120 °C. IR 3485, 2924, 1728, 1639, 1574, 1458, 1365, 1300, 1225, 1192, 1070, 993, 905 cm^–1^; ^1^H NMR (500 MHz, CDCl_3_) δ 6.01–5.88 (*m*, 1H), 5.11–5.02 (*m*, 2H), 4.79 (*s*, 1H), 3.41 (dt, *J =* 10.0, 3 Hz, 2H), 2.34, (d, *J =* 0.5 Hz, 2H), 2.15 (s, 5H), 2.05 (s, 6H); ^13^C NMR (125 MHz, CDCl_3_) δ 169.8, 150.1, 141.8, 135.5, 127.6, 126.4, 121.7, 121.4, 116.1, 31.5, 20.7, 13.3, 12.8, 12.3; HRMS (ESI) exact mass calculated for [M + H]^+^ (C_14_H_19_O_3_) requires *m*/*z* 235.1334, found *m*/*z* 235.1329.

*3-allyl-4-hydroxy-2-methylnaphthalen-1-yl acetate* (**6d**): light yellow crystal, 55% yield, mp = 117–118 °C. IR 3485, 3441, 3063, 2976, 2924, 1726, 1634, 1595, 1574, 1454, 1389, 1364, 1290, 1233, 1080, 1047, 1015, 982, 914, 768, 737 cm^–1^; ^1^H NMR (500 MHz, CDCl_3_) δ 8.11 (d, *J =* 7.5 Hz, 1H), 7.66 (d, *J =* 7.6 Hz, 1H), 7.49–7.43 (m, 2H), 6.07–6.00 (m, 1H), 5.36 (s, 1H), 5.17 (dq, *J =* 10.1, 1.6 Hz, 1H), 5.11 (dq, *J =* 18.9, 1.7 Hz, 1H), 3.59 (dt, *J =* 5.6, 1.7 Hz, 2H), 2.47 (s, 3H), 2.25 (s, 3H); ^13^C NMR (125 MHz, CDCl_3_) δ 169.7, 147.7, 138.4, 135.0, 126.7, 126.5, 126.5, 125.3, 124.0, 121.7, 120.8, 118.2, 116.7, 31.4, 20.8, 13.5; HRMS (ESI) exact mass calculated for [M + H]^+^ (C_16_H_17_O_3_) requires *m*/*z* 257.1178, found *m*/*z* 257.1172.

### 3.4. General Procedure for Acetylation

Acetic anhydride (1 mL, 10 mmol) was added to the mixture of allylphenol (5 mmol) and pyridine (10 mL) at 0 °C under an N_2_ atmosphere. After that, 4-dimethyl pyridine (0.15 g, 1 mmol) was added as catalyst. The reaction mixture was stirred for 1 h at room temperature. After completion, the mixture was poured into water and extracted with ethyl acetate. The organic layer was washed with water, saturate aqueous CuSO_4_ and brine, dried over Na_2_SO_4_, and evaporated under vacuum. The residue was purified by column chromatography with silica gel as the stationary phase (0%–5% ethyl acetate/*n-*hexane) to yield the titular allyl ester.

*2-allyl-3,5,6-trimethyl-1,4-phenylene diacetate* (**7a**): light yellow crystal, 90% yield, mp = 108–109 °C. IR 3458, 3074, 2976, 2927, 1728, 1639, 1573, 1460, 1366, 1225, 1192, 1070, 993, 905, 860, 827, 706, 683 cm^–1^; ^1^H NMR (500 MHz, CDCl_3_) δ 5.81–5.76 (m, 1H), 5.00 (dq, *J =* 10.1, 1.6, 1H), 4.94 (dd, *J =* 17.1, 1H), 3.27 (bd, *J =* 39.5 Hz, 2H), 2.34 (s, 3H), 2.31 (s, 3H), 2.06 (b, 6H, 2C**H_3_**), 2.04 (s, 3H); ^13^C NMR (125 MHz, CDCl_3_) δ 169.1 (b, 2 × **C**=O), 145.8, 145.6, 134.9, 128.4, 127.9, 127.7, 127.6, 127.4, 115.7, 31.8 (b, 2**C**H_3_CO), 20.5, 13.3, 12.7; HRMS (ESI) exact mass calculated for [M + Na]^+^ (C_16_H_20_NaO_4_) requires *m*/*z* 299.1259, found *m*/*z* 299.1257.

Our above NMR data for prepared compound (**7a**) were comparable with the NMR data of 2-allyl-1,4-diacetoxy-3,5,6-trimethylbenzen in Reference [28]: ^1^H NMR (90 MHz, CDCl_3_) δ 5.7–5.9 (m, 1H), 5.0 (d, *J* = 11, 1 H), 4.94 (d, *J* = 18, 1H), 3.28 (br s, 2H), 2.32 and 2.34 (s, 6H), 2,04 and 2,06 (s, 9H); ^13^C NMR δ 168.90, 168.70, 145.97, 145.72, 134.97, 128.47, 127.92, 127.53, 127.36, 115.4, 31.78, 20.42, 20.29, 13.07, 12.47.

*2-allyl-3-methylnaphthalene-1,4-diyl diacetate* (**7d**): brown solid, 90% yield, mp = 107–108 °C. IR 3069, 2982, 2928, 2870, 1753, 1639, 1599, 1427, 1354, 1198, 1169, 1092, 1049, 1011, 895, 773, 742 cm^–1^; ^1^H NMR (500 MHz, CDCl_3_) *δ* 7.72–7.71 (m, 1H), 7.67–7.65 (m, 1H), 7.48–7.46 (m, 2H), 5.91–5.86 (m, 1H), 5.06 (dq, *J =* 10.2, 1.6, 1H), 4.99 (dq, *J =* 17.1, 1.7, 1H), 3.48 (b, 2H), 2.49 (s, 3H), 2.47 (s, 3H), 2.27 (s, 3H); ^13^C NMR (125 MHz, CDCl_3_) δ 169.6 (2 × **C**=O) 142.9, 142.8, 134.7, 128.4, 127.1, 126.7, 126.6, 126.5, 126.3, 121.6, 121.4, 116.3, 32.1, 20.9, 20.8, 13.2. HRMS (ESI) exact mass calculated for [M + H]^+^ (C_18_H_19_O_4_) requires *m*/*z* 299.1283, found *m*/*z* 299.1282.

### 3.5. General Procedure for Stille Coupling

Bromoaryl (1.0 mmol), (PPh_3_)_2_PdCl_2_ (0.025 mmol) allylSnBu_3_ (1.1 mmol), and PPh_3_ (0.1 mmol) were dissolved in DMF (10 mL), and then the mixture was heated at 110 °C for 4 h under an N_2_ atmosphere. The reaction mixture was diluted with H_2_O, extracted five times with Et_2_O, and the combined extracts were dried over Na_2_SO_4_ and evaporated under vacuum. The residue was purified by column chromatography with silica gel as the stationary phase (10%–20% ethyl acetate/*n-*hexane) to yield the titular allylated arene.

*1-allyl-2,5-dimethoxy-3,4,6-trimethylbenzene* (**7b**): clear oil, 75% yield. IR (film) 2994, 2936, 2845, 1638, 1558, 1454, 1383, 1223, 1084, 1038, 1002, 937, 837, 758 cm^–1^; ^1^H NMR (500 MHz, CDCl_3_) δ 5.97–5.92 (m, 1H), 5.00 (dq, *J =* 10.0, 1.9 Hz, 1H), 4.90 (dq, *J =* 17.5, 2.0 Hz, 1H), 3.66 (s, 3H), 3.64 (s, 3H), 3.43 (dt, *J =* 4.0 Hz, 1.7 Hz, 2H), 2.19 (s, 9H); ^13^C NMR (125 MHz, CDCl_3_) δ 153.2, 153.0, 136.9, 129.3, 128.9, 128.2, 128.1, 114.9, 61.2, 60.3, 31.4, 12.9, 12.8, 12.0; HRMS (ESI) exact mass calculated for [M + H]^+^ (C_14_H_21_O_2_) requires *m*/*z* 221.1542, found *m*/*z* 221.1537.

*1-allyl-2,3,4,5-tetramethoxy-6-methylbenzene* (**7c**): clear oil, 75% yield. IR (film) 2934, 2829, 1728, 1638, 1464, 1404, 1350, 1292, 1257, 1196, 1105, 1036, 1009, 978, 908, 871, 743 cm^–1^; ^1^H NMR (500 MHz, CDCl_3_) δ 5.93–5.86 (m, 1H), 5.00 (dd, *J =* 10.1, 1.4 Hz, 1H), 4.91 (dd, *J =* 17.1, 1.6 Hz, 1H), 3.91 (s, 3H), 3.90 (s, 3H), 3.80 (s, 3H), 3.78 (s, 3H), 3.38 (d, *J =* 5.8 Hz, 2H), 2.14 (s, 3H); ^13^C NMR (125 MHz, CDCl_3_) δ 147.9 (2 × **C**-OMe), 145.4, 144.8, 136.7 (–**C**H=CH_2_), 127.0, 125.8, 114.9 (=**C**H_2_), 61.4, 61.3, 61.2, 60.8, 31.0, 11.7; HRMS (ESI) exact mass calculated for [M + H]^+^ (C_14_H_21_O_4_) requires *m*/*z* 253.1440, found *m*/*z* 253.1436.

*2-allyl-1,4-dimethoxy-3-methylnaphthalene* (**7e**): off-white crystal, 78% yield, mp = 83–84 °C. ^1^H NMR (500 MHz, CDCl_3_) δ 8.09–8.04 (m, 2H), 7.49–7.45 (m, 2H), 6.08–6.00 (m, 1H), 5.05 (dq, *J =* 10.2, 1.8 Hz, 1H), 4.91 (dq, *J =* 17.2, 1.9 Hz, 1H), 3.90 (s, 3H), 3.87 (s, 3H), 3.64 (dt, *J =* 5.5, 1.9 Hz, 2H), 2.39 (s, 3H); ^13^C NMR (125 MHz, CDCl_3_) δ 150.3, 150.2, 136.7, 128.6, 127.9, 127.3, 125.7, 125.4, 122.5, 122.2, 115.2, 62.5, 61.5, 31.4, 12.4; HRMS (ESI) exact mass calculated for [M + H]^+^ (C_16_H_19_O_2_) requires *m*/*z* 243.1385, found *m*/*z* 243.1380.

Our above NMR data for prepared compound (**7e**) were comparable with the NMR data of compound **5** in Reference [29]: ^1^H NMR (80 MHz, CDCl_3_) δ 8.40–8.25 (m, 2H), 7.82–7.76 (m, 2H), 6.50–6.00 (m, 1H), 5.40–4.98 (m, 2H), 4.11 (s, 3H), 4.07 (s, 3H), 3.85 (d, *J* = 6, 2H), 2.62 (s, 3H).

### 3.6. General Procedure for Key Cross Metathesis Reaction

Methyl methacrylate (200.0 mg, 2.0 mmol) and allylarene (0.1 mmol) were dissolvent in dichloromethane (100 mL), and Grubbs 2nd (42.1 mg, 0.05 mmol) was added. The reaction mixture was refluxed for 24 h at 40 °C under an N_2_ atmosphere. The mixture was then concentrated under vacuum and the residue was purified by column chromatography with silica gel as the stationary phase (5%–10% ethyl acetate/*n-*hexane) to yield the titular key ester intermediate. The *E*- configuration of the cross-coupling products was determined by NOESY NMR spectroscopy analysis.

*(E)-2-(4-methoxy-3-methyl-4-oxobut-2-en-1-yl)-3,5,6-trimethyl-1,4-phenylene diacetate* (**8a**): clear oil, 56% yield. IR (film) 2927, 2926, 1740, 1639, 1495, 1445, 1373, 1238, 1045, 935, 878, 841, 731 cm^–1^; ^1^H NMR (500 MHz, CDCl_3_) *δ* 6.58 (td, *J =* 6.8, 1.4 Hz, 1H), 3.69 (s, 3H), 3.36 (b, 2H), 2.35 (s, 3H), 2.31 (s, 3H), 2.06 (s, 3H), 2.04 (s, 6H), 1.96 (d, *J =* 1.1 Hz, 3H); ^13^C NMR (125 MHz, CDCl_3_) δ 168.4, 168.2, 146.1, 145.7, 139.5, 128.7, 128.3, 127.9, 127.4, 112.9, 51.9, 27.6, 20.7, 20.6, 13.4 (d), 13.0, 12.8; HRMS (ESI) exact mass calculated for [M + H]^+^ (C_19_H_25_O_6_) requires *m*/*z* 349.1651, found *m*/*z* 349.1646.

*Methyl (E)-4-(2,5-dimethoxy-3,4,6-trimethylphenyl)-2-methylbut-2-enoate* (**8b**): clear oil, 61% yield. IR (film) 2916, 2849, 1719, 1649, 1458, 1402, 1300, 1259, 1244, 1206, 1132, 1080, 1059, 997, 802, 743 cm^–1^; ^1^H NMR (500 MHz, CDCl_3_) δ 6.69 (td, *J =* 7.0, 1.0 Hz, 1H), 3.69 (s, 3H), 3.64 (s, 3H), 3.63 (s, 3H), 3.53 (d, *J =* 6.9 Hz, 2H), 2.19 (s, 6H), 2.17 (s, 3H), 2.01 (s, 3H); ^13^C NMR (125 MHz, CDCl_3_) δ 168.7, 153.3, 152.9, 141.3, 129.3, 129.1, 128.3, 127.9, 127.5, 61.0, 60.3, 51.8 27.1, 12.9, 12.8, 12.7, 12.5; HRMS (ESI) exact mass calculated for [M + H]^+^ (C_17_H_25_O_4_) requires *m*/*z* 293.1753, found *m*/*z* 295.1750.

*Methyl (E)-2-methyl-4-(2,3,4,5-tetramethoxy-6-methylphenyl)but-2-enoate* (**8c**): clear oil, 58% yield. IR (film) 2918, 2849, 1719, 1649, 1458, 1400, 1300, 1259, 1244, 1206, 1132, 1080, 1059, 999, 802, 741 cm^–1^; ^1^H NMR (500 MHz, CDCl_3_) *δ* 6.64 (t, *J =* 6.9 Hz, 1H), 3.91 (s, 3H), 3.89 (s, 3H), 3.79 (s, 3H), 3.78 (s, 3H), 3.70 (s, 3H), 3.48 (d, *J =* 7.0 Hz, 2H), 2.13 (s, 2H), 2.00 (s, 3H); ^13^C NMR (125 MHz, CDCl_3_) δ 168.7, 148.1, 147.9, 145.8, 144.9, 140.8, 127.5, 126.5, 125.6, 61.3, 61.2, 61.1, 60.9, 51.9, 29.8, 12.7, 12.0; HRMS (ESI) exact mass calculated for [M + H]^+^ (C_17_H_25_O_6_) requires *m*/*z* 325.1651, found *m*/*z* 325.1547.

*(E)-2-(4-methoxy-3-methyl-4-oxobut-2-en-1-yl)-3-methylnaphthalene-1,4-diyl diacetate* (**8d**): clear oil, 64% yield. IR (film) 2981, 2907, 1736, 1699, 1597, 1446, 1371, 1300, 1234, 1157, 1097, 1043, 937, 846, 785 cm^–1^; ^1^H NMR (500 MHz, CDCl_3_) δ 7.73–7.71 (m, 1H), 7.69–7.67 (m, 1H), 7.51–7.47 (m, 2H), 6.67 (td, *J =* 6.8, 1.4 Hz, 1H), 3.69 (s, 3H), 3.58 (b, 2H), 2.49 (s, 3H), 2.47 (s, 3H), 2.24 (s, 3H), 2.02 (d, *J =* 1.1 Hz, 3H); ^13^C NMR (125 MHz, CDCl_3_) *δ* 169.4, 168.3 (2 × **C=**OCH_3_), 142.9, 142.8, 138.9, 128.6, 128.1, 126.9, 126.8, 126.7, 126.5, 126.3, 121.6, 121.4, 51.9, 27.8, 20.8, 20.7, 13.4, 12.9; HRMS (ESI) exact mass calculated for [M + H]^+^ (C_21_H_23_O_6_) requires *m*/*z* 371.1495, found *m*/*z* 371.1489.

*Methyl (E)-4-(1,4-dimethoxy-3-methylnaphthalen-2-yl)-2-methylbut-2-enoate* (**8e**): clear oil, 64% yield. IR (film) 2982, 2939, 1736, 1699, 1597, 1447, 1371, 1300, 1234, 1098, 1043, 937, 847, 785 cm^–1^; ^1^H NMR (500 MHz, CDCl_3_) δ 8.07–8.03 (m, 2H), 7.50–7.46 (m, 2H), 6.74 (t, *J =* 6.8 Hz, 1H), 3.88 (s, 3H), 3.87 (s, 3H), 3.73 (d, *J =* 6.9 Hz, 2H), 3.70 (s, 3H), 2.36 (s, 3H), 2.07 (s, 3H); ^13^C NMR (125 MHz, CDCl_3_) *δ* 168.6, 150.4, 150.3, 140.7, 128.4, 128.1, 128.0, 127.3, 126.5, 125.9, 125.7, 122.4, 122.3, 62.4, 61.6, 51.9, 27.7, 12.9, 12.8; HRMS (ESI) exact mass calculated for [M + H]^+^ (C_19_H_23_O_4_) requires *m*/*z* 315.1596, found *m*/*z* 315.1592.

### 3.7. General Procedure for the Reduction of Cross Metathesis Products

To a stirred solution of ester **6c** (325.1 mg, 6.0 mmol) in anhydrous diethyl ether (10 mL) was added dropwise a suspension of LiAlH_4_ in dry THF at 0 °C. The reaction was allowed to warm to room temperature until completion as indicated by TLC (4 h). The reaction mixture was quenched at 0 °C by addition of water (2 mL) and aq sodium hydroxide (20%, 1 mL). The workup was continued by the further addition of water (10 mL) and aq sodium hydroxide (20%, 10 mL). The mixture was stirred for 30 min, after which the mineral solid precipitate was filtrated and washed with diethyl ether (40 mL). The residue was purified by column chromatography with silica gel as the stationary phase (5–10% ethyl acetate/*n-*hexane) to yield alcohol **4** (266.6 mg, 90% yield) as a clear and colorless viscous oil.

*(E)-4-(2,5-dimethoxy-3,4,6-trimethylcyclohexa-1,5-dienyl)-2-methylbut-2-en-1-ol* (**2b**)**:** clear oil, 92%. IR (film) 3435, 2954, 2927, 2870, 1724, 1641, 1456, 1400, 1375, 1248, 1148, 1082, 1006, 804, 765 cm^–1^; ^1^H NMR Spectrum (500 MHz, CDCl_3_): 5.36 (t, *J =* 6 Hz, 1H), 4.00 (s, 1H), 3.65 (s, 3H), 3.64 (s, 3H), 3.41 (d, *J* = 6 Hz, 2H), 2.19 (s, 3H), 2.18 (s, 6H); ^13^C NMR Spectrum (125 MHz, CDCl_3_): 153.29, 152.92, 134.92, 130.93, 128.79, 128.21, 127.79, 125.22, 69.10, 61.01, 60.30, 26,08, 14.10, 12.98, 12.89, 12.43; HRMS (ESI) exact mass calculated for [M + H]^+^ (C_16_H_25_O_3_) requires *m*/*z* 265.1804, found *m*/*z* 265.1802.

*(E)-2-methyl-4-(2,3,4,5-tetramethoxy-6-methylphenyl)but-2-en-1-ol* (**2c**): clear oil, 90% yield. IR (film) 3414, 2931, 2860, 1463, 1406, 1350, 1257, 1193, 1103, 1050, 1005, 970, 871 cm^–1^; ^1^H NMR (500 MHz, CDCl_3_) δ 5.34 (t, *J* 6.7 Hz, 1H), 4.00 (s, 2H), 3.90 (s, 3H), 3.90 (s, 3H), 3.80 (s, 3H), 3.78 (s, 3H), 3.37 (d, *J =* 6.7 Hz, 2H); 2.14 (s, 2H), 1.83 (d, *J =* 1.3 Hz, 3H). ^13^C NMR (125 MHz, CDCl_3_) δ 148.0, 147.8, 145.3, 144.8, 134.8, 128.4, 125.2, 124.6, 69.0, 61.3, 61.2 (b, 2**C**H_3_O), 60.7, 25.6, 14.0, 11.9; HRMS (ESI) exact mass calculated for [M + H]^+^ (C_16_H_25_O_5_) requires *m*/*z* 297.1702, found *m*/*z* 297.1701.

Our above IR and NMR data for prepared alcohol (**2c**) were comparable with the IR and NMR data of alcohol **6** in Reference [25]: IR (CHCl_3_) 3432, 2935, 2862, 1466, 1407, 1351, 1219, 1105, 1067, 1039, 1013 cm^–1^; ^1^H NMR (400 MHz, CDCl_3_) δ 5.26 (1H, m), 3.93 (2H, s), 3.83 (3H, s), 3.82 (3H, s), 3.72 (3H, s), 3.70 (3H, s), 3.29 (2H, dd, *J =* 6.8, 0.8 Hz), 2.07 (3H, s), 1.76 (3H, s), ^13^C NMR (100 MHz, CDCl_3_) δ 147.93, 147.72, 145.19, 144.75, 134.68, 128.29, 125.23, 124.52, 68.83, 61.02, 60.99, 60.63, 25.51, 13.81, 11.76.

*(E)-4-(1,4-dimethoxy-3-methylnaphthalen-2-yl)-2-methylbut-2-en-1-ol* (**2e**)**:** clear oil, 91%. IR 3397, 1659, 1597, 1454, 1379, 1366, 1263, 1159, 1066, 1037, 1007, 899, 758, 739 cm^–1^; ^1^H NMR Spectrum (500 MHz, CDCl_3_): 8.05 (m, 2H), 7.59–7.36 (m, 2H), 5.54–5.31 (m, 1H), 4.02 (d, *J* = 5.1 Hz, 2H), 3.89 (s, 3H), 3.87 (s, 3H), 3.61 (dd, *J* = 6.6, 2 Hz), 2.38 (s, 3H), 1.89 (d, *J* = 0.9 Hz, 3H); ^13^C NMR Spectrum (125 MHz, CDCl_3_): 150.42, 150.10, 135.55, 130.21, 127.83, 127.45, 126.82, 125.73, 125.56, 124.61, 122.47, 122.36, 68.98, 62.35, 61.52, 26.22, 14.19, 12.69**;** HRMS (ESI) exact mass calculated for [M + H]^+^ (C_18_H_23_O_3_) requires *m*/*z* 287.1647, found *m*/*z* 287.1645.

Our above NMR data for prepared ancol (**2e**) were comparable with the NMR data of compound **21** in Reference [30]: ^1^H NMR (400 MHz, CDCl_3_) δ 8.07–8.02 (m, 2H), 7.48–7.44 (m, 2H), 5.43–5.41 (m, 1H), 4.02 (s, 2H), 3.89 (s, 3H), 3.87 (s, 3H), 3.61 (d, *J* = 6.5, 2H), 2.38 (s, 3H), 1.90 (d, *J* = 1.0, 3H)

### 3.8. Bromination of ***2b***

Alcohol **2b** (5 mmol) was dissolved in 20 mL of anhydrous tetrahydrofuran and PBr_3_ (2.5 mmol) was slowly added to the reaction at −5 °C for 15 min under an N_2_ atmosphere. After 2 h, the reaction mixture was poured into cold saturated sodium bicarbonate solution and then extracted with ethyl acetate. The organic layer was washed with brine, dried over Na_2_SO_4_, and concentrated under vacuum. The crude bromide **9** was used for the next reaction without purification.

### 3.9. General Procedure for the Sulfination

Bromide (**9**) (4 mmol) and sodium benzene sulphinate were dissolved in 25 mL dimethyl formamide. The reaction mixture was stirred at room temperature for 12 h under an N_2_ atmosphere. The mixture was then poured into ice-cold water, acidified to pH ~4 with 2N HCl and extracted with ethyl acetate. The organic layer was washed with brine, dried over Na_2_SO_4_, and concentrated under vacuum. The residue was purified by column chromatography with silica gel as the stationary phase (0%–10% ethyl acetate/*n*-hexane) to yield the titular sulfonyl compound (**10**).

*(E)-1,4-dimethoxy-2,3,5-trimethyl-6-(3-methyl-4-(phenylsulfonyl)but-2-enyl)benzene* (**10**): white solid, 90% yield, mp = 186–187 °C**.** IR 3065, 2938, 2847, 1587, 1447, 1400, 1350, 1302, 1292, 1248, 1165, 1132, 1082, 1055, 1009, 955, 901, 768, 743 cm^–1^; ^1^H NMR Spectrum (500 MHz, CDCl_3_): 7.83–7.75 (m, 2H), 7.53 (t, *J* = 7.4 Hz, 1H), 7.42 (t, *J* = 7.8 Hz, 1H), 4.97 (t, *J* = 6.4 Hz, 1H), 3.71 (s, 2H), 3.61 (s, 3H), 3.54 (s, 3H), 3.27 (d, *J* = 6.6 Hz, 1H), 2.17 (s, 3H), 2.15 (s, 3H), 2.01 (s, 3H), 1.95 (s, 3H); ^13^C NMR Spectrum (125 MHz, CDCl_3_): 138.49, 138.46, 136.71, 135.36, 135.12, 133.50, 129.00, 128.54, 128.19, 127.85, 127.57, 123.44, 66.23, 60.93, 60.29, 26.78, 17.16, 12.96, 12.87, 12.37; HRMS (ESI) exact mass calculated for [M + H]^+^ (C_22_H_29_O_4_S) requires *m*/*z* 389.1787, found *m*/*z* 389.1784.

### 3.10. General Procedure for the Solanesylation: 

Sulfonyl (**10**) (0.50 mmol) and solanesol bromide (0.75 mmol) were dissolved in 10 mL of anhydrous tetrahydrofuran. Solution of potassium *tert*-butoxide (0.75 mmol) in 5 mL anhydrous tetrahydrofuran was added dropwise to the above solution containing sulfonyl and solanesol bromide at −25 °C. After 2 h, the reaction was allowed to reach room temperature and the stirring was continued for 12 h under an N_2_ atmosphere. After completion, the mixture was poured into ice-water, acidified to pH ~4 with 2N HCl, and extracted with ethyl acetate. The organic layer was washed with brine, dried over Na_2_SO_4_, and concentrated under vacuum. The residue was purified by column chromatography with silica gel as the stationary phase (0%–5% ethyl acetate/*n*-hexane) to yield the titular compound.

*1-((2E,6E,10E,14E,18E,22E,26E,30E,34E)-3,7,11,15,19,23,27,31,35,39-decamethyl-4-(phenylsulfonyl)tetraconta-2,6,10,14,18,22,26,30,34,38-decaenyl)-2,5-dimethoxy-3,4,6-trimethylbenzene* (**11**): clear oil, 43%. IR (film) 3296, 3233, 2914, 2849, 1728, 1468, 1418, 1290, 1217, 1194, 1177, 1101, 1047, 991, 943, 719 cm^–1^; ^1^H NMR Spectrum (500 MHz, CDCl_3_): 7.77 (m, 2H), 7.52 (t, *J* = 7.4 Hz, 1H), 7.42 (t, *J* = 7.7 Hz, 2H), 5.13–5.10 (m, 7H), 5.05 (t, *J* = 6.6 Hz, 1H), 4.97 (t, *J* = 5.9 Hz, 1H), 4.87 (t, *J* = 6.8 Hz, 1H), 3.60 (s, 3H), 3.48 (m, 4H), 3.30 (dd, *J* = 15.7, 6.9 Hz, 1H), 3.17 (dd, *J* = 15.4, 5.3 Hz, 1H), 2.86–2.77 (m, 1H), 2.68–2.56 (m, 1H), 2.16 (s, 3H), 2.14 (s, 3H), 2.10–2.04 (m, 16H), 2.02–1.92 (m, 19H), 1.84 (s, 3H), 1.68 (s, 3H), 1.64–1.50 (s, 32H), 1.26 (s, 6H); ^13^C NMR Spectrum (125 MHz, CDCl_3_): 153.16, 152.68, 135.53, 135.09, 131.41, 130.04, 128.93, 128.86, 128.86,128.13, 127.68, 126.97, 124.63, 124.50, 124.36, 123.96, 118.84, 74.07, 60.89, 60.26, 39.91, 29.89, 26.77, 26.59, 25.87, 17.84, 16.50, 16.22, 12.93, 12.87, 12.31; HRMS (ESI) exact mass calculated for [M + H]^+^ (C_27_H_37_O_4_S) requires *m*/*z* 457.2413, found *m*/*z* 457.2413.

### 3.11. General Procedure for the De-Sulfination

To a solution of **11** (0.1 mmol) in 5 mL anhydrous tetrahydrofuran was added ethanol (0.005 mol). Sodium (1.5 mmol) pieces ware slowly added to the reaction mixture under vigorous stirring at room temperature for 12 h under an N_2_ atmosphere. The reaction was then poured into ice-water and extracted with ethyl acetate. The organic layer was washed with brine, dried over Na_2_SO_4_, and evaporated under vacuum. The residue was purified by column chromatography with silica gel as the stationary phase (0%–2% ethyl acetate/*n-*hexane) to yield the titular compound.

*1-((2E,6E,10E,14E,18E,22E,26E,30E,34E)-3,7,11,15,19,23,27,31,35,39-decamethyltetraconta-2,6,10,14,18,22,26,30,34,38-decaenyl)-2,5-dimethoxy-3,4,6-trimethylbenzene* (**12**): clear oil, 41%. IR (film) 3103, 2989, 2936, 2626, 1587, 1510, 1450, 1400, 1300, 1246, 1169, 1130, 1246, 1169, 1130, 1082, 1053, 1002, 903, 822, 741 cm^–1^; ^1^H NMR Spectrum (500 MHz, CDCl_3_): 5.28–5.00 (m, 10H), 3.64 (s, 6H), 3.36 (d, *J* = 6.3 Hz, 2H), 2.69 (m, 3H), 2.18 (s, 12H), 2.08(m, 22H), 2.06(m, 24H), 1.78 (s, 3H), 1.68 (s, 6H), 2.63–1.56 (m, 38H); ^13^C NMR Spectrum (125 MHz, CDCl_3_): 153.25, 152.90, 135.27, 135.22, 135.13, 135.08, 131.84, 128.43, 128.07, 127.94, 124.62, 124.46, 124.33, 123.46, 123.36, 61.06, 60.29, 40.38, 39.94, 39.89, 27.21, 26.97, 26.92, 26.82, 26.37, 25.88, 16.50, 16.21, 12.96, 12.89, 12.34, 12.21; HRMS (ESI) exact mass calculated for [M + H]^+^ (C_21_H_33_O_2_) requires *m*/*z* 317.2481, found *m*/*z* 317.2478.

## 4. Conclusions

In conclusion, we devised an alternative synthesis route of benzylallyl ester intermediates for the preparation of coenzyme Q_10_ and its derivatives. The key features of this strategy were the use of catalytic formation of allylarenes and cross-metathesis reactions to selectively prepare *E*-benzylallyl esters. The resultant esters were reduced to benzylallylic alcohols on the pathway toward coenzyme Q_10_ and derivatives. One alcohol was utilized to synthesize a coenzyme Q_10_ derivative.

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
