# Peer review of "Preparation of Key Intermediates for the Syntheses of Coenzyme Q10 and Derivatives by Cross-Metathesis Reactions"

_molecules, 2020, doi:10.3390/molecules25030448_

Round 1
Reviewer 1 Report
The manuscript entitled ‘Preparation of Key Intermediates for the Synthesis of Coenzyme Q10 and Derivatives by Cross-metathesis Reactions’
by Trang Nguyen, Hung Mac and Phong Pham
reports the synthesis, catalyzed by 2nd gen. Grubb’s catalyst, of five benzylmethacrylate esters from allylarenes and methylmethacrylate. Then, they carried out the reduction of some products to alcohols and reported one example of functionalization to give a derivative bearing a solanesyl moiety. The authors claim to have synthesized Coenzyme Q10 (in the title) and derivatives, but did not performed the deprotection step to obtain Coenzyme Q10.
The manuscript is not particularly innovative since other papers reported a similar approaches to obtain Coenzyme Q10 and its derivatives. Moreover, a careful revision is needed prior to be considered for publication. Some points are as follows:
Deprotection of 10 to give quinol derivative must be done
In Scheme 1 the first structure must be indicated as quinol or hydroquinone, or benzene-1,4-diol
In Scheme 2 the numbering of compounds must be added
In Scheme 3, the first step to obtain compounds 5 require nirogen athmosfere: thi must be indicated
Table 1: footnotes b and c are the same
Scheme 4. Why only 6b, c, e have been subjected to reduction?
Experimental: Compounds derived from General procedure for allylation reaction must be numbered and the path used must be indicated
Page 5, refeences 1 and 2 are inserted as footnotes: references must numbered consecutively and inserted in the dedicated section.
Reference 12: the first part is a note, not a bibliography: it must be inserted in the main text.
Supportin information: many spectra indicated as 13C seem to be spectra obtained from DEPT experiments: 13C NMR spectra must be added.
Some other typos are present along the text (e.g: schem, inseperable, etc)
Author Response
Dear Reviewer:
Thank you very much for your comments on our manuscript. We have fixed our manuscript as your detail suggestions as being presented in the following table.
|
Comments |
Responds |
1 |
Deprotection of 10 (now 12) to give quinol derivative must be done |
This is one derivative of coenzyme Q10 among others in our library for the biological evaluation phase. We have initially tried similar deprotection with CAN but got quite messy mixture. We will scale up and optimize this reaction and apply to our library. |
2 |
In Scheme 1 the first structure must be indicated as quinol or hydroquinone, or benzene-1,4-diol |
We have fixed the name. |
3 |
In Scheme 2 the numbering of compounds must be added |
Compounds in Scheme 2 have been added numbers. |
4 |
In Scheme 3, the first step to obtain compounds 5 require nirogen athmosfere: this must be indicated |
Reaction’s condition under nitrogen atmosphere have been added to the experimental section. |
5 |
Table 1: footnotes b and c are the same |
Condition b: using 5.0 equiv. of methylmethacrylate, not 20 equiv. We have fixed the footnotes. |
6 |
Sheme 4. Why only 6b, c, e have been subjected to reduction? |
Compounds 6a (8a) and 6d (8d) were used as selected cross-metathesis examples. Compound 6c (8c) was reduced to form common alcohol 2c for the formal synthesis of coenzyme Q10. Compounds 6b (8b), and 6e (8e) were reduced to prove the same concept of similar syntheses. |
7 |
Experimental: Compounds derived from General procedure for allylation reaction must be numbered and the path used must be indicated |
We have fixed this. |
8 |
Page 5, refeences 1 and 2 are inserted as footnotes: references must numbered consecutively and inserted in the dedicated section. |
References have been added to the references list. |
9 |
Reference 12: the first part is a note, not a bibliography: it must be inserted in the main text. |
We have modified and added in the main text. |
10 |
Supportin information: many spectra indicated as 13C seem to be spectra obtained from DEPT experiments: 13C NMR spectra must be added. |
13C NMR spectra were recorded in JMOD mode which are the combination of 13C and DEPT without losing quarternary 13C signals. |
11 |
Some other typos are present along the text (e.g: schem, inseperable, etc) |
We have checked and fixed typos. |
Thank you again for your suggestion.
Your sincerely,
Phong Pham, Hung Mac, and Trang Nguyen
Reviewer 2 Report
Pham et al. describe a new approach for the synthesis of coenzyme Q10 and some other derivatives, there is not much to say about the MS but that the chemistry is sound and seem to have been performed in a proper manner with the conclusion reached in line with the results presented. I believe that a careful revision of the English grammar would certainly improve the MS. Once this query has been properly addressed, I believe the MS can be accepted for publication.
Author Response
Dear Reviewer:
Thank you very much for your kindly support to our manuscript.
Your sincerely,
Phong Pham, Hung Mac, and Trang Nguyen
Reviewer 3 Report
The paper describes the preparation of key intermediates for the synthesis of coenzyme Q10 and derivatives by cross-metathesis reactions. The work is interesting, well performed and nicely described. The product characterization is adequate. The results are important to scientists in the field of synthetic chemistry.
I suggest that the paper is accepted after minor revisions.
Requested changes:
-The images in the schemes look fuzzy and of low quality. Please improve.
-In page 2, section ''design plan'' the authors mention compound numbers 2, 3 and 4 but those are not included in the following scheme which makes it confusing to the reader. Please add the compound numbering under each structure in scheme 2.
-The authors claim to have saved one synthetic step by starting from the methoxy ether of starting aryl bromide instead of the phenol. This is not really the case as the methoxy ether is probably a derivative of the phenol thereby the extra step exists but was performed by the industry. One needs to consider these details when claiming to improve important synthetic routes.
-Use of Stille coupling is best to be avoided due to the toxicity of stannyl reagents. Have the authors considered alternative Suzuki protocols?
-The structure of 2c in scheme 5 is rotated differently to all the previous schemes and table. This can be confusing to the reader. Please rotate all structures the same way.
Regarding the experimental:
-Compounds 4a, 4e and 2e are reported in the literature. Please add a reference to the literature proceedures in the experimental and compare your data to the ones reported. This should be done for all literature compounds.
-The authors need to report IR data for all new compounds.
Author Response
Dear Reviewer:
Thank you very much for your kindly comments on our manuscript. We have fixed our manuscript as your detail suggestions as being presented in the following table.
|
Comments |
Responds |
1 |
The images in the schemes look fuzzy and of low quality. Please improve. |
It seems that our first version did not work well on Mac. We have used Window to prepare this version. |
2 |
In page 2, section ''design plan'' the authors mention compound numbers 2, 3 and 4 but those are not included in the following scheme which makes it confusing to the reader. Please add the compound numbering under each structure in scheme 2. |
Compounds in Scheme 2 have been added numbers. |
3 |
The authors claim to have saved one synthetic step by starting from the methoxy ether of starting aryl bromide instead of the phenol. This is not really the case as the methoxy ether is probably a derivative of the phenol thereby the extra step exists but was performed by the industry. One needs to consider these details when claiming to improve important synthetic routes. |
We propose to use more catalytic approach only as an option (alternatively) to avoid repeatedly stoichiometric oxidation/reduction sequences when using free quinol and unfunctionalized iso-pentenyl group as starting materials. Also, this approach may be apply in medicinal chemistry lab scale for preparing library of coenzyme Q10’s derivatives. |
4 |
Use of Stille coupling is best to be avoided due to the toxicity of stannyl reagents. Have the authors considered alternative Suzuki protocols? |
Suzuki protocol is good alternative to this coupling reaction, but we used only Stille because of the availability of chemicals in the laboratory. |
5 |
The structure of 2c in scheme 5 is rotated differently to all the previous schemes and table. This can be confusing to the reader. Please rotate all structures the same way. |
We have fixed the Schemes for the unification. |
6 |
Compounds 4a (7a), 4e (7e) and 2e are reported in the literature. Please add a reference to the literature proceedures in the experimental and compare your data to the ones reported. This should be done for all literature compounds. |
We have added a references to the reference’s list and compared your data to the reported compounds and they are quite comparable. |
7 |
The authors need to report IR data for all new compounds. |
IR spectra have been added both to the main text and SI. |
Thank you again for your suggestion.
Your sincerely,
Phong Pham, Hung Mac, and Trang Nguyen
Round 2
Reviewer 1 Report
The manuscript, after the revision, is suitable for publication.